# Electronic distance learning satisfaction, awareness, attitudes, and barriers among undergraduate nursing students following the october 7 war in Palestine

**Ashraf Jehad Abuejheisheh**[1]*, **Rabia H. Haddad**[2], **Wafaa J. Tohol**[3], **Salam Bani Hani**[4]

**1** Assistant Professor, Nursing Department, Faculty of Health Professions, Al-Quds University, Jerusalem-Palestine, **2** Assistant Professor, Faculty of Nursing, Philadelphia University-Amman-Jordan, **3** School of Nursing, The University of Jordan-Amman-Jordan, **4** Assistant Professor, Nursing Department, Irbid National University, Irbid, Jordan

* ageisha@staff.alquds.edu

## Abstract

### Background

Education is one of the many facets of daily life that have been severely interrupted by the ongoing conflict in Palestine since October 7, 2023. Assessing the efficacy and difficulties of electronic distance learning (EDL) requires an understanding of undergraduate nursing students' experiences with it during this period.

### Aim

The study aimed to assess the levels of satisfaction, awareness, attitudes, preferences, and barriers towards EDL distance learning among undergraduate nursing students during the war.

### Methods

A descriptive cross-sectional design was used. A valid and reliable self-administered questionnaire was utilized to assess the levels of satisfaction, awareness, attitudes, preferences, and barriers toward EDL among undergraduate nursing students.

### Results

A total of 292 nursing students participated in this cross-sectional study. Participants demonstrated an overall positive attitude toward EDL (M = 3.02, SD = 0.529), particularly regarding flexibility and ease of use. However, overall satisfaction with EDL was low (M = 2.84, SD = 0.885), revealing a clear contrast between favorable perceptions and actual learning satisfaction. Most students (80.9%) were aware of EDL. Infrastructural barriers were the most prominent challenges, with electricity shortages

**Data availability statement:** Data cannot be made public to protect student privacy, as per regulations imposed by the Department of Nursing at Al-Quds University. Data requests can be sent to Dr. Farid Ghrayeb, the Head of the nursing Department of Nursing at Al-Quds University via email at fghrayeb@staff.alquds.edu.

**Funding:** The author(s) received no specific funding for this work.

**Competing interests:** The authors have declared that no competing interests exist.

(64.7%), high internet costs (62.7%), and poor connectivity (61.6%) dominating the EDL experience. Satisfaction levels differed significantly by sex, age group, and digital tool used; however, causal inferences are limited due to the cross-sectional design.

## Conclusion

The study's findings highlight how crucial it is to continue planning and coming up with solutions that both fill in the gaps in EDL and capitalize on its benefits. The educational institution decision-makers should give this issue a top priority and create a blended learning system to improve communication between instructors and students. Taking into account the security of students during pandemics and wars.

## Introduction

Over 75 million children and youth in 35 crisis-affected countries require educational support [1]. **Electronic Distance Learning (**EDL) has the potential to help close the educational gap, but it must be carefully planned, implemented, and evaluated [2]. According to an EDL statistics report by Nikou, Kim [3]. The global EDL industry was estimated to be worth $250 billion in 2020 and is expected to increase at a compound annual growth rate of 21% from 2021 to 2027.

It was predicted that there would be 1.6 billion EDL users worldwide in 2020, and that number would rise to 2.5 billion by 2025 [4]. An interdisciplinary and practical study topic, EDL in the context of the war in Palestine after October 7, 2023, pulls from a variety of disciplines, including education, computer science, psychology, sociology, nursing, and peace studies [5]. For instance, a study performed by Aqtam, Naghnaghiyeh [6] pointed to a profound mental health emergency among undergraduate students in the West Bank who have been exposed to conflict. Urgent action is required, including the implementation of trauma-focused cognitive behavioral therapy adapted to the realities of ongoing conflict, alongside school-based mental health initiatives. Additionally, it encompasses a variety of stakeholders with varying responsibilities and interests in EDLin conflict areas, including researchers, nursing students, nursing educators, university administrators, policymakers, practitioners, and humanitarian workers [7]. Therefore, EDL in light of the war after October 7, 2023, in Palestine, requires a comprehensive and collaborative approach that considers the diverse perspectives and realities of the people involved in EDL in Palestine [8]. Additionally, a research topic that examines the benefits and difficulties of implementing online education in conflict-affected areas is Electronic Learning (E-learning) in War. It seeks to comprehend how EDL might contribute to the continuity and caliber of education for teachers and students who encounter several hazards and obstacles [8].

Face-to-face instruction was once the norm, but as technology advanced, EDL became necessary [9]. Online or in-person blended learning has become popular as a result of certain events, like the coronavirus and war [10]. Numerous advantages, including adaptability, accessibility, and customization, can be obtained through e-learning. The digital gap, technical network constraints, a lack of proficiency in e-learning, a lack of freedom, poor time management, workload pressure, and language barriers are just a few of the difficulties that come with this collaboration [11].

Palestine is a relevant and timely research topic, as it sheds light on the impact of the conflict on the education sector and the resilience of the students and educators who have been affected by it [12]. According to Watch [13], the war that began on October 7, 2023, has resulted in unprecedented levels of civilian casualties, displacement, and humanitarian crisis in both Israel and Palestine, with Gaza being the most severely affected area. The war has also disrupted the normal functioning of schools and universities, forcing many of them to close or operate under limited capacity.

Although EDL offers numerous advantages in war-affected areas, there are also challenges to consider, namely the lack of infrastructure, resources, policies, and skills [14]. According to a recent study by Li and Saleh [15], teachers in Palestine faced numerous obstacles to EDL implementation during the COVID-19 pandemic, but they also agreed on ways to overcome them. The report suggested that classrooms should be outfitted with Internet-connected computers and that a strategy for emergency schooling be created. In order to prepare instructors, students, and school administrators to use the EDL resources, the research also invited educational authorities to provide training.

This study sheds light on awareness, attitudes, preferences, and barriers regarding EDL among undergraduate nursing students in Palestine during times of conflicts like war may hold significant implications for understanding the flexibility and effectiveness of education like EDL in challenging environments. Thus, enabling the development of tailored interventions to overcome obstacles and ensure uninterrupted access to quality education. By addressing these factors, educators and policymakers, such as administrators and Information Technology (IT) specialists in Al-Quds University, can strengthen the adaptability and inclusivity of EDL initiatives, thereby empowering the newly graduated nursing students and ensuring that the nursing graduates have a high level of competency in terms of the nursing profession. Despite the growing global reliance on e-learning, empirical evidence on its effectiveness and student experiences in active conflict settings remains scarce. Existing studies on digital learning are largely derived from stable or post-conflict contexts, where infrastructural, psychological, and security conditions differ substantially from those experienced during ongoing warfare [16]. This study addresses a critical gap in the literature by examining nursing students' attitudes, satisfaction, awareness, and barriers toward EDL during an active conflict, where electricity disruptions, internet instability, and safety concerns fundamentally shape the learning experience. By capturing students' lived educational realities under war conditions, the findings provide context-specific evidence that moves beyond technological readiness to highlight the structural and humanitarian dimensions of educational continuity in conflict zones.

In conflict areas, EDL offers a chance to get past obstacles to education and equip people with skills and information. Even in times of instability, EDL can help ensure that education continues and offers hope for a better future by utilizing technology. Thus, this study aims to evaluate undergraduate nursing students' levels of satisfaction, awareness, attitudes, preferences, and barriers regarding EDL platforms during times of war. The study hypothesized that there are no statistically significant differences at $\alpha \leq 0.05$ in the level of awareness, attitudes, preferences, and barriers among nurse students at Al-Quds University according to sex, age, place of residency, academic year, Cumulative GPA, family income, and employee status.

This study will answer the following research questions:

1. What is the level of satisfaction, attitude, and awareness of EDL among nursing students at Al-Quds University?

2. Are there any significant differences in the level of satisfaction, level of awareness, and attitude between nursing students using EDL based on demographics?

## Methods

### Design

A descriptive cross-sectional design was used to assess nursing students' satisfaction, awareness, attitudes, and barriers regarding EDL following the October 7 war in Palestine. This design was suitable because it allowed the collection of data from a large group at a single point in time, providing a snapshot of students' experiences during a unique post-conflict period. It also enables the identification of associations between key variables while accommodating practical constraints such as limited mobility and safety restrictions [17].

### Participants and sampling

This study employed a cross-sectional design and was conducted among undergraduate nursing students enrolled in the Faculty of Health Professions, Nursing Department at Al-Quds University, Palestine, during the period following the October 7, 2023, war. The target population comprised all registered nursing students across the four academic years at the university. At the time of data collection, the accessible population included 1,146 students who were eligible to participate.

Eligibility criteria included: [1] enrollment as an undergraduate nursing student at Al-Quds University during the study period, and [2] willingness to participate voluntarily. Students who were not enrolled during the data collection period or who submitted incomplete questionnaires were excluded from the analysis.

A non-probability convenience sampling approach was used due to the constraints imposed by the ongoing conflict, including mobility restrictions and safety concerns. All eligible students (N = 1,146) were invited to participate via official WhatsApp student groups, which represent the primary and most reliable communication channel between the faculty and students during the war. The survey link remained active for an extended period, and reminders were sent to enhance participation. Sample size was calculated using G*Power software, indicating that a minimum of 288 participants was required to detect a moderate effect size (0.30) with a significance level of $\alpha = 0.05$ and a statistical power of 0.80 [18]. A total of 292 students completed the questionnaire, slightly exceeding the required sample size to account for potential attrition and missing data. This resulted in a response rate of approximately 25.5%.

### Data collection tools

A self-administered questionnaire was used to assess undergraduate nursing students' satisfaction, awareness, attitudes, preferences, and perceived barriers toward e-learning during the war in Palestine. The instrument was adapted from previously validated tools developed by Olum and Atulinda [16] and Nikou and Maslov [17]. Adaptations involved contextual and linguistic modifications to ensure relevance to a conflict-affected educational environment, including rewording items to reflect war-related challenges such as internet instability, electricity outages, safety concerns, and displacement, as well as alignment with locally used e-learning platforms. No new domains were introduced; rather, existing items were refined to maintain conceptual consistency with the original instruments.

Content validity was reassessed by a panel of nursing education academics and e-learning specialists, who evaluated item relevance, clarity, and appropriateness for the study objectives and context. Based on their feedback, minor revisions were made to improve wording and eliminate redundancy. Construct validity was examined using factor analysis, which confirmed appropriate item clustering under five domains: satisfaction (10 items), awareness of e-learning tools and methods (8 items), attitudes toward e-learning (12 items), preferences for e-learning modalities (7 items), and perceived barriers to e-learning (9 items).

A pilot study was conducted to reassess the reliability of the adapted instrument, yielding acceptable internal consistency (Cronbach's $\alpha = 0.79$ for attitudes and 0.83 for satisfaction). Most items were measured using a 5-point Likert scale ranging from strongly disagree to strongly agree, with selected yes/no questions for categorical responses. The questionnaire was administered online using the WhatsApp application.

## Ethical considerations

Institutional Review Board approval was secured from the assigned University in compliance with the research protocol for nursing students in Palestine, Ref No: (ERC-2024–37). The study followed the ethical principles of the World Medical Association's Declaration of Helsinki. The data collected from the students in the online survey were anonymous, and no identifiers or personal information were collected or stored. Informed consent and a statement explaining the purpose of the study and the participants' rights regarding voluntary and confidential participation were included in the email invitation to students. A digital consent form was provided at the start of the questionnaire because the study was conducted online (via WhatsApp). Participants were required to click "I agree" or provide a written "yes" response before proceeding to the survey. The participants were assured that data would be used only for research purposes and that their privacy and confidentiality would be protected.

## Data collection procedure

After obtaining the ethical approvals from the research ethics committee at Al-Quds University, the data collected from the students in the online survey were anonymous during the period from 10/March /2024 to 25/ April /2024. Data collection methods included a questionnaire, created by researchers under the title EDL during War after October 7, 2023, in Palestine: Awareness, Attitudes, Preferences, and Barriers Among Undergraduate Nursing Students. It was mentioned on the cover page of the online instrument that all data was dealt with confidentially. The research purpose and contact details were mentioned on the cover page.

The questionnaire was distributed online on WhatsApp groups to be filled in at the time of the specified lectures. It consisted of two parts, the first one identified some general information about the participants, whereas the second part comprised 19 questions. The questionnaire was performed with 292 undergraduate nursing students, all of whom had experienced EDL for at least one year.

## Data analysis

The study investigators reviewed the questionnaires for missing and invalid answers, and those were eliminated. The response rate was calculated, and the responses were coded, tabulated, and reviewed. The collected data were analyzed by the Statistical Package for Social Sciences (SPSS) Version 27. Data entry was performed and double-checked for outliers or errors, as well as tested for normality. Data analysis of descriptive and inferential statistics was conducted. Descriptive statistics, including frequency, percentages, mean, and standard deviation (SD), were used to describe the study variables. Inferential statistics was utilized, including independent sample t-test and One-Way ANOVA, to assess the mean differences between all variables.

## Results

### Demographic characteristics

A total of 292 nursing students participated in this study. The result reveals that the majority of participants were female (67.8%), and most of the participants' ages were between 20–29 years (71.2%). In terms of residence, 47.9% lived in a city. Academically, the fourth-year students constitute the largest group of participants (36.0%). The cumulative GPA distribution indicates that a majority score is between 80–89% (53.4%). Digital tool usage is dominated by mobile phones (50.0%) and laptops (33.2%). Internet connectivity quality is perceived as moderate by most of the participants (64.4%), averaging out to a mean quality rating of (M = 3.13, SD = 0.739). Table 1

### Attitude level toward EDL

Overall, the participants showed a moderate to positive attitude toward EDL(M = 3.02, *SD* = 0.529). The pie chart, indicating generally favorable perceptions, nearly two-thirds of students (63.7%) exhibited a positive attitude (mean ≥ 3),

**Table 1. Demographic variables of the participants (N = 292).**

| Demographics | | n | % |
|---|---|---|---|
| **Sex** | Male | 94 | 32.2% |
| | Female | 198 | 67.8% |
| **Age Group** | 18-19 years old | 77 | 26.4% |
| | 20-29 years old | 208 | 71.2% |
| | >30 years old | 7 | 2.4% |
| **Place of residence** | City | 140 | 47.9% |
| | Village | 130 | 44.5% |
| | Camp | 22 | 7.5% |
| **Academic year** | 1st year | 72 | 24.7% |
| | 2nd year | 68 | 23.3% |
| | 3rd year | 47 | 16.1% |
| | 4th year | 105 | 36.0% |
| **Cumulative GPA** | 90-99% | 18 | 6.2% |
| | 80-89% | 156 | 53.4% |
| | 70-79% | 110 | 37.7% |
| | 60-69% | 8 | 2.7% |
| **Digital tool** | Laptop | 97 | 33.2% |
| | Mobile phone | 146 | 50.0% |
| | I-Pad | 40 | 13.7% |
| | Computer Disk | 9 | 3.1% |
| **Rated internet connectivity** | Very low quality | 8 | 2.7% |
| | Low quality | 27 | 9.2% |
| | Moderate quality | 188 | 64.4% |
| | High quality | 57 | 19.5% |
| | Very high quality | 12 | 4.1% |
| | **Mean: 3.13 SD:.739** | | |
| **Family income** | 1500-2000 | 40 | 13.7% |
| | 2100-3000 | 59 | 20.2% |
| | 3100-4000 | 51 | 17.5% |
| | 4100-5000 | 63 | 21.6% |
| | 1360 $>= | 79 | 27.1% |
| **Employee status** | Yes | 134 | 45.9% |
| | No | 158 | 54.1% |

while 36.3% demonstrated a negative attitude toward EDL. All mean scores were interpreted in relation to the theoretical midpoint of the 5-point Likert scale (3.0), which represents a neutral response. Accordingly, the attitude mean score of 3.02 was interpreted as marginally positive, reflecting only a slight deviation from neutrality rather than a strong positive attitude. Although students expressed positive attitudes toward the flexibility and ease of use of EDL, their perceptions of instructional quality and teaching effectiveness remained negative, reflecting a nuanced and context-dependent attitude. Please see Fig 1 below regarding attitude level of study participants toward EDL.

In Table 2, the results present that on average, participants hold a positive attitude toward e-learning. The belief that EDL should only be for distributing notes over the internet is perceived as a positive attitude among participants (M = 3.08, SD = 1.057). Participants also agree that EDL provides schedule flexibility (M = 3.10, SD = 0.979). The ease of use of EDL technology received a mean score of (M = 3.22, SD = 1.073). On the other hand, the quality of knowledge attained through

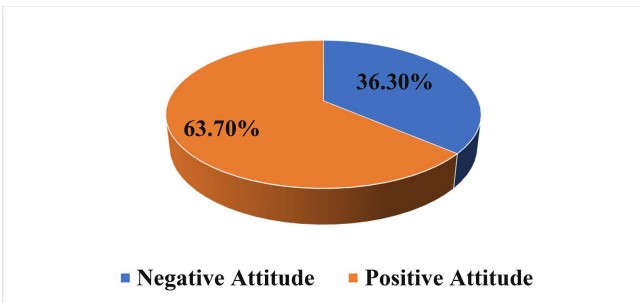

**Fig 1. Attitude level of study participants toward EDL (<3 = dissatisfied, ≥ 3 = satisfied).**

EDL is perceived negatively (M = 2.84, *SD* = 1.130), with an interpretation indicating mild dissatisfaction in relation to the theoretical midpoint of the 5-point Likert scale (3.0). Similarly, the efficiency of EDL as a teaching method is viewed negatively (M = 2.85, *SD* = 1.130). All negatively worded attitude items were reverse-coded before analysis, and the means reported represent the final post–reverse coding scores used to classify attitudes as positive or negative.

### Satisfaction level toward EDL

The overall satisfaction score indicated a low level of satisfaction with EDL among participants (M = 2.84, SD = 0.885). Students reported dissatisfaction with distance learning overall and felt that EDL did not adequately meet their learning needs, nor maintain course quality comparable to traditional instruction. Table 3

### Awareness level toward EDL

Table 4 presents awareness and preferences toward EDL among nursing students. For instance, the results revealed that 80.9% of participants were aware of e-learning. Only a small portion of the students (8.2%) never visit academic websites or other related applications. More than half of the participants(57.2%) visit academic websites or other related applications. When it comes to the need for training to use EDL platforms effectively, opinions vary; the largest proportion of students are neutral (48.3%), while 4.8% strongly agree that they would require training. Preferences for learning methods were different, with nearly half of the students (47.3%) favoring a combination of classroom lectures and e-learning. A significant number prefer conventional classroom lectures only (33.6%), and 19.2% would choose EDL exclusively.

### Barriers to EDL

A range of challenges impedes EDL access for nursing students. It is noteworthy to mention that most students had great barriers with availability and electricity (64.7%), followed by prohibitive internet costs (62.7%), poor internet connectivity (61.6%), sub-optimal work conditions (59.6%), inadequate equipment (56.2%), and insufficient EDL skills (43.8%). Table 5.

### Mean differences of attitudes toward EDL-based on demographics

One-way ANOVA, the results revealed that there was a statistically significant difference in attitude concerning age group ($F_{2, 289}$ = 3.475, p = 0.032). Post-hoc comparisons using the Bonferroni test revealed that the mean score for the age group of those between 20–29 years old (M = 3.05, *SD* = 0.532) was significantly higher than those at age of 18–19 years old years (M = 2.906, *SD* = 0.461). In addition, the results revealed that there was a statistically significant difference in attitude concerning the academic year ($F_{3, 288}$ = 3.363, p = 0.019). Post-hoc comparisons using the Bonferroni test revealed that the mean score for the third-year students (M = 3.15, *SD* = 0.638) was significantly higher than first-year students (*M* = 2.87,

**Table 2. The mean score for each item toward the attitude of EDL (N=292).**

| Item | Mean | SD | Interprertation |
|---|---|---|---|
| **Attitudes of study participants to EDL** | | | |
| EDL should only be used for the distribution of notes over the internet (R). | 3.08 | 1.057 | Positive |
| EDL ensures schedule flexibility. | 3.10 | .979 | Positive |
| EDL reduces the quality of knowledge attained (R). | 2.84 | 1.126 | Negative |
| EDL technology is easy to use. | 3.22 | 1.073 | Positive |
| EDL is not as efficient as a teaching method (R). | 2.85 | 1.130 | Negative |
| **Total Mean Score (5 items)** | **3.02** | **.529** | **Positive** |

*Mean score over 5, R: reversed coded item, (<3=negative attitude, ≥3= positive attitude).*

**Table 3. The mean score for each item toward the satisfaction of EDL(N=292).**

| Item | Mean | SD | Status |
|---|---|---|---|
| **Satisfaction of EDL among nursing students at Al-Quds University** | | | |
| I am very satisfied with distance learning. | 2.68 | 1.116 | Dissatisfied |
| I feel that EDL courses serve my needs well. | 2.92 | 1.000 | Dissatisfied |
| I feel the quality of the courses I took was largely unaffected by conducting them via the Internet. | 2.95 | 1.054 | Dissatisfied |
| I would prefer online courses to traditional courses for my studies. | 2.81 | 1.152 | Dissatisfied |
| **Total Mean Score (4 items)** | **2.84** | **.885** | **Dissatisfied** |

*Mean score over 5, (<3=dissatisfied, ≥3= satisfied).*

**Table 4. Awareness and preferences towards EDL(N=292).**

| Item | | n | % |
|---|---|---|---|
| Have you heard about e-learning? | Yes | 236 | 80.8% |
| | No | 56 | 19.2% |
| Have often do you visit academic websites or applications? | Never | 24 | 8.2% |
| | Sometimes | 167 | 57.2% |
| | Always | 101 | 34.6% |
| I would require training to use EDL platforms effectively. | Strongly Disagree | 23 | 7.9% |
| | Disagree | 48 | 16.4% |
| | Neutral | 141 | 48.3% |
| | Agree | 66 | 22.6% |
| | Strongly agree | 14 | 4.8% |
| Preferred learning methods. | Both classroom lectures and e-learning. | 138 | 47.3% |
| | Conventional classroom lectures only. | 98 | 33.6% |
| | EDL only. | 56 | 19.2% |

$SD=0.483$). However, the results revealed that attitude toward EDL was not statistically significantly different based on other demographics (e.g., sex, place of residence, Cumulative average..etc). **Table 6**.

## Mean differences of EDL satisfaction based on demographics

Overall, independent sample t-tests and one-way ANOVA revealed statistically significant differences in satisfaction scores according to sex, age group, and type of digital tool used. Male students reported significantly higher satisfaction with EDL

**Table 5. Barriers to EDL access among nursing students (N = 292).**

| Item | Yes | | No | |
|---|---|---|---|---|
| | n | % | n | % |
| Lack of gadgets (tools or equipment such as e-devices). | 164 | 56.2% | 128 | 43.8% |
| Lack of EDL skills. | 128 | 43.8% | 164 | 56.2% |
| Conditions at work are unpleasant or sometimes even unsafe. | 174 | 59.6% | 118 | 40.4% |
| Availability and Electricity. | 189 | 64.7% | 103 | 35.3% |
| Poor internet connectivity. | 180 | 61.6% | 112 | 38.4% |
| Internet costs. | 183 | 62.7% | 109 | 37.3% |

than female students. In addition, older students (20–29 years and >30 years) demonstrated higher satisfaction levels compared to younger students aged 18–19 years. Regarding digital tools, students who primarily used mobile phones or computer disks reported higher satisfaction than those who relied on laptops, suggesting that accessibility and device suitability played an important role in shaping EDL satisfaction during the war.

The results revealed that there was a statistically significant difference in satisfaction based on sex ($t_{290} = 2.036$, $p = 0.043$). Knowing that males have a higher satisfaction mean score (M = 2.99, SD = 0.949) than females (M = 2.77, SD = 0.846). Moreover, using one-way ANOVA, the results revealed that there was a statistically significant difference in satisfaction concerning the age group ($F_{2, 289} = 4.036$, p = 0.019). Post-hoc comparisons using the Bonferroni test revealed that the mean score for the age group of those between 18 and 19 years (M = 2.62, *SD* = 0.81) was significantly lower than those at the age of 22–29 years (M = 2.92, *SD* = 0.89). Finally, using one-way ANOVA, the results revealed that there was a statistically significant difference in satisfaction concerning digital tools ($F_{3, 288} = 2.484$, p = 0.023). Post-hoc comparisons using the Bonferroni test revealed that the satisfaction mean score for the mobile phone was (M = 2.95, SD = 0.865) higher than Laptop (M = 2.66, *SD* = 0.855). Moreover, Post-hoc comparisons using the Bonferroni test revealed that the satisfaction mean score for the computer disk was (M = 3.33, *SD* = 0.943) higher than the Laptop (M = 2.66, SD = 0.855). Table 7

## Discussion

The 2023 war in Palestine forced undergraduate nursing students to transition to electronic distance learning, a shift driven by necessity rather than choice. This shift examines students' satisfaction, awareness, attitudes, and barriers in a conflict-stricken context.

The observed discrepancy between students' positive attitudes toward EDL and their low satisfaction levels represents a key finding of this study. While students acknowledged the flexibility, accessibility, and ease of use of e-learning platforms, reflecting a favorable cognitive and affective attitude, these advantages did not translate into satisfaction with learning outcomes. This disconnect can be attributed to external, war-related barriers such as electricity shortages, unstable internet connectivity, unsafe learning environments, and limited access to appropriate devices. Consequently, students may value EDL in principle but remain dissatisfied in practice due to factors beyond their control, underscoring the importance of contextual and infrastructural readiness in conflict-affected educational settings.

Given the difficulties brought on by the ongoing conflict, electronic distance learning among undergraduate nursing students in Palestine after the events of October 7, 2023, is crucial. This study shed light on nursing students' satisfaction, awareness, attitudes, preferences, and barriers toward e-learning. Despite the majority of nursing students showing an overall positive attitude toward e-learning. This result was inconsistent with the findings of a study performed by Olum, Atulinda [19] who reported that overall, medical and nursing students had a negative attitude toward e-learning, as indicated by their mean attitude score. On the other hand, a cross-sectional study conducted in Nepal by Thapa and Bhandari [18] found that nearly half of nursing students expressed a favorable opinion toward e-learning, despite its use as a replacement during the COVID-19 pandemic. In the current study, a positive attitude can be interpreted in several ways.

**Table 6. Mean Differences between demographic variables of the participants in terms of the attitude mean score (N = 292).**

| Demographic variable | | Mean± SD | F/t | DF | P-value |
|---|---|---|---|---|---|
| **Sex** | Male | 3.05±.612 | | | .569 |
| | Female | 3.01±.486 | | 1 | |
| **Age Group** | 18-19 years old | 2.906±.461 | | | **.032*** |
| | 20-29 years old | 3.05±.532 | 3.475 | 2 | |
| | >30 years old | 3.31±.900 | | | |
| **Place of residence** | City | 2.98±.489 | | | .129 |
| | Village | 3.09±.585 | | 2 | |
| | Camp | 2.90±.379 | | | |
| **Academic year** | 1st year | 2.87±.483 | | | **.019*** |
| | 2nd year | 3.00±.504 | 3.363 | 3 | |
| | 3rd year | 3.15±.638 | | | |
| | 4th year | 3.08±.504 | | | |
| **Cumulative average** | 90-99% | 2.96±.554 | | | .066 |
| | 80-89% | 2.97±.517 | | 3 | |
| | 70-79% | 3.12±.535 | | | |
| | 60-69% | 2.77±.483 | | | |
| **Digital tool** | Laptop | 3.00±.542 | | | .134 |
| | Mobile phone | 3.04±.496 | | 3 | |
| | iPad | 2.92±.539 | | | |
| | Computer Disk | 3.35±.760 | | | |
| **Rated internet connectivity** | Very low quality | 3.22±.483 | | | .372 |
| | Low quality | 2.89±.563 | | | |
| | Moderate quality | 3.05±.513 | | 4 | |
| | High quality | 3.01±.544 | | | |
| | Very high quality | 2.86±.651 | | | |
| **Family income** | 400-550 $ | 3.10±.335 | | | .889 |
| | 551-700 $ | 3.00±.531 | | | |
| | 701-850 $ | 3.02±.533 | | | |
| | 851-1000 $ | 2.99±.612 | | 4 | |
| | 1001>= $ | 3.02±.543 | | | |
| **Employee status** | Yes | 3.08±.511 | | | .101 |
| | No | 2.97±.541 | | 1 | |

*Independent t-test and One-Way ANOVA, *Significant at p<0.05.*

Still, the most significant is that EDL has saved students' lives in the face of contemporary employment circumstances and prevented them from wasting time traveling to university in the presence of obstacles. Additionally, it has allowed students to save a substantial amount of money.

**Table 7. Mean Differences between demographic variables of the participants in terms of the satisfaction mean score (N = 292).**

| Demographic variable | | Mean ± SD | F/t | DF | P-value |
|---|---|---|---|---|---|
| **Sex** | Male | 2.99 ± .949 | | | **.043*** |
| | Female | 2.77 ± .846 | 2.036 | 1 | |
| **Age Group** | 18-19 years old | 2.61 ± .809 | | | **.019*** |
| | 20-29 years old | 2.91 ± .899 | | | |
| | >30 years old | 3.25 ± .866 | 4.036 | 2 | |
| **Place of residence** | City | 2.76 ± .886 | | | .257 |
| | Village | 2.88 ± .899 | | | |
| | Camp | 3.06 ± .772 | | 2 | |
| **Academic year** | 1st year | 2.72 ± .820 | | | .101 |
| | 2nd year | 2.78 ± .836 | | | |
| | 3rd year | 3.12 ± 1.006 | | | |
| | 4th year | 2.84 ± .888 | | 3 | |
| **Cumulative average** | 90-99% | 2.93 ± .906 | | | .485 |
| | 80-89% | 2.76 ± .874 | | | |
| | 70-79% | 2.93 ± .909 | | | |
| | 60-69% | 2.93 ± .716 | | 3 | |
| **Digital tool** | Laptop | 2.66 ± .855 | | | **.023*** |
| | Mobile phone | 2.95 ± .865 | | | |
| | iPad | 2.76 ± .942 | | | |
| | Computer Disk | 3.33 ± .943 | 2.484 | 3 | |
| **Rated internet connectivity** | Very low quality | 2.59 ± .972 | | | .293 |
| | Low quality | 2.61 ± .824 | | | |
| | Moderate quality | 2.92 ± .887 | | | |
| | High quality | 2.72 ± .863 | | | |
| | Very high quality | 2.91 ± .990 | | 4 | |
| **Family income** | 400-550 $ | 2.55 ± .855 | | | .210 |
| | 551-700 $ | 2.81 ± .746 | | | |
| | 701-850 $ | 2.92 ± .937 | | | |
| | 851-1000 $ | 2.96 ± .930 | | | |
| | 1001 >= $ | 2.86 ± .913 | | 4 | |
| **Employee status** | Yes | 2.93 ± .905 | | | .127 |
| | No | 2.77 ± .864 | | 1 | |

*Independent t-test and One-Way ANOVA, *Significant at p < 0.05.*

Post-October 7, 2023, nursing students in Palestine's post-conflict region may find satisfaction with EDL amidst physical destruction, curfews, and limited mobility. The EDL offers flexibility and adaptability but may face practical challenges like unreliable internet, power outages, and limited device access. This result was consistent with Farrah, Zainh [20] who reported that online learning can provide learners with opportunities to employ learning technologies during challenging

conditions to secure somewhat conducive learning environments during wartime. However, the theoretical focus of EDL may not fully replicate patient care skills, and the emotional toll of war may further diminish satisfaction.

Besides, the results revealed that most nursing students were dissatisfied with EDL in the current study. Similarly, a study conducted in India during the pandemic of COVID-19 pandemic by Kanagaraj, Sakthivel [21] reported that the majority of students expressed dissatisfaction with the technical and environmental aspects of virtual learning, although rating their satisfaction levels as "good" in virtual theory classes and "moderate" in virtual practical learning. According to these findings, EDL strategies should be developed to improve learning outcomes and guarantee a higher degree of satisfaction with online learning activities. On the other hand, a study conducted by Tayyib, Alsolami [22] reported that undergraduate nursing students who were immediately placed in an unfamiliar EDL curriculum setting expressed high levels of satisfaction and showed emerging technological proficiency, perseverance, and fortitude. This finding could be related to several reasons, including a lack of engagement in studies compared with face-to-face learning, the inability to interact with teachers in person and understand their requirements, and the poor quality of online exams. This includes issues such as short time limits for some exams and the large number of questions.

Additionally, the apparent paradox between marginally positive attitudes toward EDL and reported dissatisfaction can be theoretically understood as a form of pragmatic acceptance shaped by crisis conditions. Nursing students may perceive EDL positively because it offers a critical lifeline for maintaining academic progression, particularly through schedule flexibility and the ability to continue learning despite mobility restrictions. However, this pragmatic endorsement does not equate to satisfaction with the quality of EDL implementation. Dissatisfaction likely reflects the inability of virtual learning to replicate essential clinical training, limited interaction with instructors, and the cumulative stressors associated with war-related disruptions. Thus, positive attitudes toward EDL may coexist with dissatisfaction, representing adaptive coping rather than genuine approval of the learning modality.

Most nursing students in the current study were aware of the learning requirements and principles. This data suggests a high level of awareness of EDL among nursing students, with varying levels of engagement and differing opinions on the need for additional training to use EDL platforms effectively. There's a clear preference for a blended learning approach, combining traditional classroom instruction with e-learning. This result was inconsistent with a study performed by Karaaslan, Çelik [23] that reported the unpreparedness of nursing students regarding distance learning. They stated that the provision of basic data, economic situation, degree of technological software competence, degree of technological device proficiency, and asynchronous learning were the main factors that affect the success of distance learning. In the current study, nursing students could limit their engagement in the effective use of EDL due to limited access to technology and insufficient training. Therefore, it is recommended that digital literacy be included in the nursing curriculum. Besides, the awareness of EDL among Palestinian nursing students varies due to prior exposure to technology and institutional support. The COVID-19 pandemic and conflict may have prepared students for EDL transitions. However, limited funding and resources may have hindered effective communication, and socioeconomic disparities may influence students' understanding of EDL processes.

In the current study, nursing students' preferences for learning methods are varied, with nearly half of the participants favoring a combination of classroom lectures and EDL, since they perceived a range of challenges that impede EDL access for nursing students—the most notable issues with availability and electricity, prohibitive internet costs, and poor internet connectivity.

Students' attitudes towards EDL in the context of the war are likely a mix of pragmatism, frustration, and ambivalence [24]. While the ability to continue education may foster resilience, negative attitudes may arise due to perceived inadequacy of EDL for nursing training, lack of face-to-face interaction, cultural factors, and the war's psychological impact, leading to heightened stress and uncertainty [25]. Besides, post-war Palestine faces numerous barriers to EDL due to technological, logistical, psychological, and educational factors. These findings were consistent with the findings of a study performed by Bakar, Shah [26] who reported that infrastructure destruction, displacement, trauma, and lack of clinical

settings make it difficult for students to charge devices or maintain stable internet connections. Also, mental health challenges and inadequate institutional support further complicate the situation. The war likely worsened electricity shortages in Gaza, making it difficult for students to focus on learning.

Addressing these infrastructural and educational deficiencies is crucial to enhancing EDL accessibility. These findings were consistent with studies performed by Thapa, Bhandari [27] and Khatatbeh, Amer [28] who stated that most of the nursing students experienced technical difficulties and internet troubles. EDL has the potential to be a crucial alternative teaching technique and learning tool in the nursing sector if it can be made user-friendly with fewer technological obstacles and enhanced with programs that can improve practical learning abilities.

Barriers to EDL identified in this study should be understood as an extension of pre-existing challenges within nursing education in the West Bank. A study performed by Bsharat [29] documented significant structural and experiential deficiencies in clinical learning environments, including limited supervision, overcrowded training sites, and restricted hands-on opportunities. The transition to EDL during the conflict compounded these challenges by introducing new barriers such as unreliable connectivity, limited access to devices, and the absence of practical skill acquisition. Together, these findings suggest that nursing students are navigating overlapping disruptions across both clinical and virtual learning environments, intensifying educational inequities during the conflict.

To sum up, the findings of this study must be interpreted within the exceptional educational and psychosocial conditions that followed the events of October 7, 2023, in the West Bank. Unlike disruptions observed during the COVID-19 pandemic, the shift to EDL occurred amid active political violence, repeated displacement, electricity shortages, internet instability, and pervasive fear and uncertainty. These conditions fundamentally shaped students' engagement with EDL, transforming it from a pedagogical choice into a survival mechanism for educational continuity. ***Implications***

The educational institution decision-makers should give this issue a top priority and create a blended learning system to improve communication between instructors and students. Taking into account the security of students during pandemics and wars. Students will develop critical technical skills for utilizing online platforms and managing contemporary technology through this method, which will also enhance comprehension and promote the sharing of ideas. Additionally, to help teachers improve their skills in implementing blended learning, it also suggests more training programs. This implementation enables educational institutions to be well-equipped to handle similar situations in the future, including pandemics or emergency closures.

For policy implications, the findings indicate that electricity shortages and poor internet connectivity are significant barriers impacting the effectiveness of EDL during conflicts, leading to student dissatisfaction despite overall positive attitudes. To tackle these issues, university administrators should implement contingency academic plans that include flexible scheduling, asynchronous learning, and low-bandwidth materials. IT specialists should focus on optimizing learning platforms for mobile use, providing offline resources, and establishing technical support for unstable connectivity. Furthermore, external humanitarian organizations can enhance infrastructure resilience by offering subsidized internet access, power backup solutions, and digital devices for vulnerable students, thus fostering collaboration between universities and humanitarian partners to maintain nursing education quality in conflict settings.

## Strengths and limitations

Despite the dominance of technology in the educational field and its significant reliance on it, especially amidst the ongoing conflict in Palestine, this is the first study conducted to assess the satisfaction, awareness, attitudes, preferences, and barriers to EDL in universities in Palestine. However, the ongoing conflict in Palestine may limit researchers' access to a diverse sample of nursing students, potentially underrepresenting those in heavily impacted areas like Gaza and skewing results towards less-affected regions. Besides, the use of electronic surveys or interviews during the war may introduce bias, with power outages potentially compromising data quality. For more clarification, the use of WhatsApp for data collection, while pragmatic in the study context, may have introduced selection bias by potentially excluding students

experiencing the most severe internet or connectivity disruptions. Besides, the study's timing after the war may lead to temporal bias, exaggerating negative attitudes towards EDL, making it difficult to distinguish between transient reactions and stable perceptions. Additionally, self-reporting bias in student satisfaction, awareness, attitudes, and barriers can introduce subjectivity, as students may overstate or understate experiences due to social desirability, fear, and cultural norms. Finally, multiple inferential comparisons were conducted without formal adjustment for multiplicity, which may increase the likelihood of Type I error. Therefore, statistically significant findings should be interpreted cautiously and confirmed in future studies using more conservative analytical approaches.

## Conclusion

In conclusion, this study provides valuable evidence of the complex and often paradoxical perceptions of EDL among nursing students in a war-affected region. By situating the findings within the specific psychological, educational, and infrastructural realities of post–October 7 Palestine and drawing on robust regional scholarship, the study contributes meaningfully to the emerging literature on nursing education in conflict settings. The results highlight the need for context-sensitive, trauma-informed educational models that move beyond technological access to address the broader human dimensions of learning under political violence. In order to ensure that nursing education and the future healthcare workforce can endure the strains of conflict, addressing these issues calls for both academic innovation and more extensive humanitarian support.

## Author contributions

**Conceptualization:** Ashraf Jehad Abuejheisheh, Rabia H. Haddad.

**Data curation:** Ashraf Jehad Abuejheisheh, Wafaa J. Tohol, Salam Bani Hani.

**Formal analysis:** Ashraf Jehad Abuejheisheh, Rabia H. Haddad.

**Investigation:** Rabia H. Haddad, Wafaa J. Tohol, Salam Bani Hani.

**Methodology:** Rabia H. Haddad, Wafaa J. Tohol.

**Project administration:** Ashraf Jehad Abuejheisheh.

**Resources:** Salam Bani Hani.

**Supervision:** Ashraf Jehad Abuejheisheh.

**Validation:** Rabia H. Haddad, Wafaa J. Tohol, Salam Bani Hani.

**Visualization:** Rabia H. Haddad.

**Writing – original draft:** Ashraf Jehad Abuejheisheh.

**Writing – review & editing:** Rabia H. Haddad, Wafaa J. Tohol, Salam Bani Hani.

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
