## [Decision Letter · Decision Letter 0]

10 Dec 2025

Dear Dr. Abuejheisheh,

Thank you for submitting your manuscript to PLOS ONE. After careful consideration, we feel that it has merit but does not fully meet PLOS ONE’s publication criteria as it currently stands. Therefore, we invite you to submit a revised version of the manuscript that addresses the points raised during the review process.

We look forward to receiving your revised manuscript.

Kind regards,

Ahmad H. Al-Nawafleh, Ph.D, MPA, CI, RN

Academic Editor

PLOS One

Journal Requirements:

3. In the online submission form, you indicated that the datasets used and/or analyzed during the current study are available from the corresponding author on reasonable request.

5. We note that there is identifying data in the Supporting Information file <RESC Dr. Ashraf Abuejheisheh Distance Learning.pdf>. Due to the inclusion of these potentially identifying data, we have removed this file from your file inventory. Prior to sharing human research participant data, authors should consult with an ethics committee to ensure data are shared in accordance with participant consent and all applicable local laws.

-Location data

Additional Editor Comments:

Dear authors,

Thank you for your effort in this manuscript. I hope you can work on the comments provided by the reviewers.

Looking forward to see your reply shortly.

All the best

Reviewers' comments:

Reviewer's Responses to Questions

**Comments to the Author**

1. Is the manuscript technically sound, and do the data support the conclusions?

Reviewer #1: Yes

Reviewer #2: Partly

Reviewer #3: Partly

Reviewer #4: Yes

Reviewer #5: Yes

2. Has the statistical analysis been performed appropriately and rigorously?

Reviewer #1: Yes

Reviewer #2: Yes

Reviewer #3: No

Reviewer #4: Yes

Reviewer #5: Yes

3. Have the authors made all data underlying the findings in their manuscript fully available?

Reviewer #1: Yes

Reviewer #2: Yes

Reviewer #3: No

Reviewer #4: Yes

Reviewer #5: Yes

4. Is the manuscript presented in an intelligible fashion and written in standard English?

Reviewer #1: Yes

Reviewer #2: Yes

Reviewer #3: Yes

Reviewer #4: Yes

Reviewer #5: Yes

Reviewer #1: This is a timely and highly relevant study. The research addresses a critical, under-researched area the impact of conflict on educational delivery, specifically for nursing students. The descriptive cross-sectional design is appropriate for assessing awareness, attitudes, and barriers. The clarity of the abstract and the summary of key findings are strong.

Areas of Strength

•Timeliness and Relevance: The study is extremely relevant as it focuses on the educational disruptions immediately following the October 7, 2023, conflict in Palestine, providing valuable insights into crisis-time education.

•Clear Objectives and Design: The Aim is clearly stated (assessing satisfaction, awareness, attitudes, preferences, and barriers). The Descriptive Cross-Sectional Design is suitable for gathering baseline data on these variables.

•Strong Results Section: The abstract clearly presents the major quantitative findings, including the overall positive attitude (M = 3.02) contrasted with the overall dissatisfaction (M = 2.84), which is a compelling result.

•Identification of Key Barriers: The study successfully isolated concrete, practical barriers directly related to the conflict/infrastructure (electricity/availability, internet costs, connectivity), providing actionable data for policymakers.

•Adequate Sample Size: The sample size calculation (N=288 required, N=292 recruited) demonstrates statistical rigor and addresses the need for adequate power.

Areas for Improvement and Further Detail

1. Clarity and Consistency of Terminology

•E-learning vs. EDL: The manuscript uses "Electronic Distance Learning (EDL)" and "e-learning" interchangeably. While the context is clear, consistent use of a single term (e.g., EDL) throughout the manuscript (title, abstract, methods) is essential for precision.

2. Methods Section Enhancements

•Detailed Tool Description: While the abstract mentions a "valid and reliable self-administered questionnaire," the methods section needs more detail. Specifically:

oSource of Adaptation: Clarify how the instruments from Olum, Atulinda (16) and Nikou and Maslov (17) were adjusted and whether these adjustments necessitated a new pilot for validity.

3. Data Presentation

•Statistical Significance Context: The abstract mentions, "The results showed significant differences in satisfaction scores related to sex, age group, and the type of digital tool used." This is a key finding that should be explained more in the results section, including the direction of the differences (e.g., were females more satisfied? Were mobile users less satisfied than laptop users?).

•Attitude vs. Satisfaction Disconnect: The mean scores need to be consistently presented as high, medium, or low. The contrast between a positive attitude (M=3.02) and dissatisfaction (M=2.84) is the most impactful finding. Ensure the discussion fully explores why students might think e-learning is flexible (positive attitude) but still be dissatisfied (due to the external barriers).

4. Discussion and Conclusion

•Blended Learning Justification: The conclusion recommends a "blended learning system." The discussion should explicitly link the specific barriers (electricity, internet) to this recommendation.

•Policy Implications: Strengthen the direct link between the key barriers (electricity/internet) and specific, actionable recommendations for university administrators, IT specialists, and external humanitarian organizations.

Reviewer #2: This is a valuable study conducted under exceptionally challenging circumstances, and its focus on EDL in a active conflict zone is critically important. However, major revisions are required to ensure methodological rigor and interpretative clarity. Please to see attached Review Report

Reviewer #3: Dear Author(s),

This research article addresses a very timely and relevant topic related to crisis conditions by examining ‘satisfaction, awareness, attitude, and barriers to e-learning among nursing students after the October 7 war in Palestine.’ The topic is highly important and can help fill the gap in the literature regarding education during crisis situations. However, the article in its current form has serious shortcomings in reporting methodology, data analysis, and discussion structure that must be addressed before publication in a reputable international journal.

Title and Abstract

The abstract presents contradictory results. In one sentence, it mentions that the overall attitude is positive, and in another sentence, it says the mean satisfaction indicates dissatisfaction. This contradiction needs clarification. It is suggested that the interpretation of results in the abstract be clearer. For example, it should state that although the overall attitude toward the idea of e-learning was positive, the practical experience of it during war conditions led to a low level of satisfaction.

Methodology

The methodology section has several fundamental weaknesses. First, the sampling method is not clearly explained. Was it random, a census, or convenience sampling? This ambiguity casts a shadow over the generalizability of the results. Also, inclusion and exclusion criteria for eligible students are not mentioned, and the response rate is not reported. Given the distribution through WhatsApp, mentioning the number of invitations sent and the number of respondents is critical for evaluating the likelihood of bias. Therefore, the ‘Participants and Sampling’ section should be completely rewritten and include precise details of the sampling method, inclusion and exclusion criteria, and response rate.

Another weakness is the lack of operational definitions for variables. How exactly were the concepts of ‘satisfaction,’ ‘attitude,’ and ‘awareness’ measured and with what indicators? A complete list of items for each construct should preferably be provided in an appendix. Additionally, the scoring method and interpretation of the Likert scale, for example, the reason for choosing a cutoff of 3 for positive or negative, needs justification. It is suggested to add a separate section titled ‘Variables and Measurement’ that clearly defines each construct and how it was measured. Regarding the validity and reliability of the instrument, although reliability is mentioned, the validity of the instrument is not adequately reported. How was the ‘adaptation’ process of the instrument carried out? Was exploratory or confirmatory factor analysis conducted to examine the structure of the constructs? Merely mentioning review by an expert panel is not sufficient. Providing details of the content and construct validation process is necessary.

Findings

The reporting of findings is incomplete. For qualitative variables such as awareness and preferences, only percentages are reported, and it would be better to include the number alongside the percentage in the text. In tables related to mean differences, the exact P value for all comparisons is not mentioned and is only shown as an asterisk, which is unprofessional. Exact p-values should be reported. Also, for post-hoc tests after ANOVA, only the existence of a difference is mentioned, and reporting adjusted P values for each pairwise comparison is necessary. Therefore, the findings should be reported with greater precision, and n values, exact p values, and results of post-hoc tests should be included. Furthermore, greater caution is needed in interpreting the mean attitude, which is very close to the neutral cutoff point and is considered ‘positive.’ Can this result really be called ‘positive’ or ‘neutral to slightly positive’? It is suggested to be more cautious in interpreting this finding and to mention the limitation of using a fixed cutoff point.

Discussion

The discussion section lacks logical structure. Instead of coherent interpretation of findings within the framework of research objectives, comparisons with other studies are made in a scattered manner. The interpretation is incomplete; for example, why is ‘satisfaction’ low despite ‘overall positive attitude’? This apparent contradiction should be analyzed more deeply in the discussion, with reference to identified barriers and the difference between attitude toward the ‘idea’ of e-learning and the practical ‘experience’ of it during war conditions. Some comparisons with studies conducted in completely different conditions may be misleading. Also, for some statistically significant differences such as higher satisfaction among men or mobile users, no explanation or hypothesis is provided. It is suggested that the discussion section undergo fundamental rewriting. The suggested structure could include a summary of key findings, interpretation of main findings in light of barriers and war conditions, selective comparison with studies in similar contexts, possible explanation for demographic-based differences, emphasis on practical implications, and greater clarity regarding limitations.

Limitations

Although some limitations are mentioned, the inherent design limitations of cross-sectional studies are not mentioned. This study does not show a causal relationship. Additionally, using an online questionnaire during electricity and internet outages could lead to severe selection bias in favor of individuals who were in days or areas with better access. It is suggested that these key limitations be clearly included in the limitations section and their potential impact on results be discussed.

Minor Issues

There is inconsistency in references. In the text, two different sources are cited with the same number, indicating an error in the order of sources. Also, the statement regarding data availability that states ‘data are available from the corresponding author upon reasonable request’ may not be sufficient for journals that emphasize open data access. Efforts to upload data to a public repository are recommended.

Final Summary

This article has high potential for publication because it addresses a critical and understudied topic. However, at present, due to serious shortcomings in reporting methodology, analysis, and interpretation of results, it is not at the level of a reputable international journal. My recommendation is MAJOR REVISION.

If the authors can completely address the mentioned weaknesses, especially in the methodology, reporting of findings, and discussion structure sections, the article will significantly improve and substantially increase the chance of final acceptance.

Reviewer #4: I would like to thank you for the invitation to review the manuscript entitled “Electronic Distance Learning Satisfaction, Awareness, Attitudes, and Barriers among Undergraduate Nursing Students Following the October 7 War in Palestine ”.

Overall, this is an excellent piece of work that will add significant value to the body of knowledge regarding strategies, such as nursing education and the impact of war on the learning process. It is a well-written manuscript, and I recommend its minor revision. However, I suggest that minor points should be addressed.

Methods:

•Describe in further detail why you selected this research design and why it is suitable for your study.

•Add Detailed inclusion/exclusion criteria in separate subheadings.

Discussion:

•The implications section is rather short and could be expanded upon by incorporating some of the relevant material appearing elsewhere in the paper.

•Please make the limitations section clearer and highlight more of your study's strengths.

Reviewer #5: Thank you for the invitation to review the manuscript entitled “Electronic Distance Learning Satisfaction, Awareness, Attitudes, and Barriers among Undergraduate Nursing Students Following the October 7 War in Palestine ”.

Thank you for the pivotal and imperative study that is needed in the such conflict zone. The manuscript is well-structured and methodologically sound, but requires minor clarifications. It is a well-written manuscript, and I recommend its acceptance.

The following points to strengthen the work:

1-The methods and data provide a solid foundation for assessing the impact of Electronic Distance Learning Satisfaction, Awareness, Attitudes, and Barriers among Undergraduate Nursing Students. Explain more why you chose this design and why this Research design is appropriate for your study.

2- The manuscript provides a comprehensive and balanced assessment of existing studies on Electronic Distance Learning.

3- Enhance discussion by incorporating some studies related to the context of Palestinian situation anf factors that may influcne distance learning as well as methods to overcome obstacles that face nursing students in the conflict country.

4- In the result section: What was the final response rate? And how many participants were initially invited?

5- A language revision is required throughout the manuscript to improve readability and clarity.

6- Addressing the suggested revisions will significantly enhance the scientific value and reproducibility of the study.

**Do you want your identity to be public for this peer review?** For information about this choice, including consent withdrawal, please see our For information about this choice, including consent withdrawal, please see our Privacy Policy .

Reviewer #1: No

Reviewer #2: No

Reviewer #3: No

Reviewer #4: No

Reviewer #5: No

---

## [Author Response · Author response to Decision Letter 1]

7 Jan 2026

I would like to thank you for handling the manuscript entitled "Electronic Distance Learning Satisfaction, Awareness, Attitudes, and Barriers among Undergraduate Nursing Students Following the October 7 War in Palestine". I would like to inform you that we have taken into consideration all the comments. I upload a word file namely "R1 author response" in the attached files.

---

## [Decision Letter · Decision Letter 1]

16 Mar 2026

Dear Dr. Ashraf Jehad Abuejheisheh,

Thank you for submitting your manuscript to PLOS ONE. After careful consideration, we feel that it has merit but does not fully meet PLOS ONE’s publication criteria as it currently stands. Therefore, we invite you to submit a revised version of the manuscript that addresses the points raised during the review process.

**ACADEMIC EDITOR: Please insert comments here and delete this placeholder text when finished.** Be sure to:

Indicate which changes you require for acceptance versus which changes you recommend

Address any conflicts between the reviews so that it's clear which advice the authors should follow

Provide specific feedback from your evaluation of the manuscript

publication criteria  and not, for example, on novelty or perceived impact. and not, for example, on novelty or perceived impact.

We look forward to receiving your revised manuscript.

Kind regards,

Ahmad H. Al-Nawafleh, Ph.D, MPA, CI, RN

Academic Editor

PLOS One

Journal Requirements:

Additional Editor Comments (if provided):

Dear Authors,

Thank you for your efforts in amending and responding to the reviewers' comments.  Please recheck the language and fine-tune your sentences before submission.  I suggest you work on the AI writing parts too for your interest. I understand that you may have used it for paraphrasing.

All the best,

Reviewers' comments:

<!----comment node--<!--a=1

---

## [Editor Report · Acceptance letter]

PONE-D-25-61436R1

PLOS One

Dear Dr. Abuejheisheh,

I'm pleased to inform you that your manuscript has been deemed suitable for publication in PLOS One. Congratulations! Your manuscript is now being handed over to our production team.

Kind regards,

on behalf of

Pro Fadwa Alhalaiqa

Academic Editor

PLOS One